# Non-Cell Autonomous and Epigenetic Mechanisms of Huntington’s Disease

**DOI:** 10.3390/ijms222212499

**Published:** 2021-11-19

**Authors:** Chaebin Kim, Ali Yousefian-Jazi, Seung-Hye Choi, Inyoung Chang, Junghee Lee, Hoon Ryu

**Affiliations:** 1Brain Science Institute, Korea Institute of Science and Technology, Seoul 02792, Korea; cbkim@kist.re.kr (C.K.); yousefian@kist.re.kr (A.Y.-J.); shchoi323@kist.re.kr (S.-H.C.); 2Department of Biology, Boston University, Boston, MA 02215, USA; ic313@bu.edu; 3Boston University Alzheimer’s Disease Research Center, Boston University, Boston, MA 02118, USA; 4Department of Neurology, Boston University School of Medicine, Boston, MA 02118, USA; 5VA Boston Healthcare System, Boston, MA 02130, USA

**Keywords:** Huntington’s disease, non-cell autonomous pathway, astrocyte, oligodendrocyte, epigenetics, mitochondria dysfunction, vesicle trafficking, therapeutic targets

## Abstract

Huntington’s disease (HD) is a rare neurodegenerative disorder caused by an expansion of CAG trinucleotide repeat located in the exon 1 of *Huntingtin (HTT)* gene in human chromosome 4. The HTT protein is ubiquitously expressed in the brain. Specifically, mutant HTT (mHTT) protein-mediated toxicity leads to a dramatic degeneration of the striatum among many regions of the brain. HD symptoms exhibit a major involuntary movement followed by cognitive and psychiatric dysfunctions. In this review, we address the conventional role of wild type HTT (wtHTT) and how mHTT protein disrupts the function of medium spiny neurons (MSNs). We also discuss how mHTT modulates epigenetic modifications and transcriptional pathways in MSNs. In addition, we define how non-cell autonomous pathways lead to damage and death of MSNs under HD pathological conditions. Lastly, we overview therapeutic approaches for HD. Together, understanding of precise neuropathological mechanisms of HD may improve therapeutic approaches to treat the onset and progression of HD.

## 1. Introduction

Huntington’s disease (HD) is a fatal progressive neurodegenerative disorder with a mid-life onset which ranges from infancy to the ninth decade. HD occurs in 5–10 cases per 100,000 persons worldwide, and characterized by chorea, emotional distress, and progressive cognitive decline [1]. Generally, in HD patients, there are more than 38 repeats of trinucleotide CAG within the *Huntingtin (HTT)* gene with an inverse relationship between the number of CAG repeats and the age of onset, indicating that the high number of CAG repeats cause the earlier phenotype of HD symptoms [2,3,4]. HTT has a crucial role in embryonic development by showing the death of embryos of huntingtin homozygous knockout mice by day 7.5 [5]. On the other hand, HTT plays an important role in cardiomyocytes cellular energy and nucleotides metabolism [6].

Currently, HD is considered as a multi-systemic neurodegenerative disease due to skeletal muscle and heart function disturbances with energy metabolism and mitochondrial alterations beyond the brain dysfunction [7]. Although most of studies show intracellular inclusions formed by mutant HTT (mHTT) protein in selected regions of the brain such as the striatum and cortex of HD brain [8,9], other studies find the expression and function of mHTT in skeletal muscle and heart as well [10]. The mHTT is involved in transcriptional alterations, disruption of intracellular transport, excitotoxicity, collapse of protein degradation mechanisms, mitochondrial dysfunction, and disorders of myelin, which make neurons more susceptible to generic stresses, eventually leading to neuronal death [11,12,13,14]. Yamanishi et al. have reported a novel cell death pathway known as transcriptional repression-induced atypical cell death of neurons (TRIAD) in the HD pathology, exhibiting that enlargement of endoplasmic reticulum (ER) contributes to neuronal cell damage without alteration of mitochondria and nuclei structure [15]. Another study has shown that inflammation may play a role in cardiac dysfunction in HD by overexpression of the inflammatory cytokine, tumor necrosis factor-α, in cardiomyocytes of R6/2 HD mice [16].

The epigenetic modifications are closely associated with the pathogenesis of HD [17,18]. mHTT sequesters specific transcription factors and impairs their function, resulting in disruption of their target genes transcription [19,20,21]. mHTT also regulates transcription by facilitating transcriptional factor interaction with protein complexes [22]. Otherwise, mHTT leads to histone hypoacetylation which can change genes expressions [23,24,25]. Moreover, several studies have shown that the biochemical defect and impairment of neuronal energy metabolism in HD patients are caused by mitochondrial dysfunction, where mitochondria is a major contributor of energy production and a regulator of intracellular signaling and survival [26,27]. The calcium-ion (Ca^2+^) buffering abnormalities and bioenergetic impairments in mitochondria can occur by interaction of mHTT with mitochondrial proteins [28,29,30]. Recently, researchers have become more interested in identifying the roles of non-neuronal cells in HD pathophysiology through elucidation of alterations in oligodendrocyte functions such as myelin formation [31,32,33] and astrocyte morphological changes and functions including impairment of glutamate metabolism and potassium homeostasis [34,35,36]. In this review, we discuss how astrocytes and oligodendrocytes contribute to HD pathogenesis via a non-cell autonomous pathway. Furthermore, we review the different epigenetic modifications which have key roles in neurotoxicity and neurogenesis impairment, and finally discuss potential therapeutic strategies for HD.

## 2. Non-Cell Autonomous Cell Death Pathway in HD

While the primary causes of neuronal damage and cell death are abnormalities within the damaged neuron itself, other neighboring cells such as glial cells also contribute to neuronal death [37,38]. In this regard, non-cell autonomous pathway is defined in this review as the mechanism of the neuronal damage caused by non-neuronal cells. Recent studies have proven the non-cell autonomous pathway in that reactive astrocytes lead to neuronal damage and neurodegeneration in Alzheimer’s disease (AD), amyotrophic lateral sclerosis (ALS), HD, and Parkinson’s disease (PD) [39,40,41,42]. Indeed, neuron-specific expression of mHTT in the striatum and cortex is not sufficient to fully induce pathological phenotypes of HD [43]. Otherwise, astrocyte-specific expression of mHTT does not induce neurodegeneration, but it shows neurological symptoms [44]. In the HD fly model (UAS HttQ100-mRFP) where mHTT is expressed in neurons, suppression of mHTT aggregation specifically in astrocytes expands the lifespan of HD fly [45]. On the other hand, white matter is degenerated and the oligodendrocyte differentiation is defective in HD [46]. mHTT-expressing microglia are hyperreactive to inflammatory stimuli to cause synaptic dysfunction in dendritic spines [47,48,49], where neurons try to control microglia activation but fail in HD [50]. Both mHTT and wtHTT are secreted from brain cells [51], where mHTT oligomers disrupt energy metabolism of neighboring cells [52]. mHTT not only affects brain metabolism, it also causes the dysfunction of other organs such as liver [53]. In the below sections, we discuss how mHTT affects astrocytes and oligodendrocytes in the neuropathogenesis of HD.

### 2.1. Alteration of Astrocyte Function

Astrocytes support neural functions by managing local environments including nutrients, ions, and neurotransmitters. In addition to the role as a generic supporter of neurons, astrocytes are involved in information processing [54], especially in the form of tripartite synapse [55]. Within the tripartite synapse, astrocytes tightly envelop synapses to clear residual neurotransmitters and ions after synaptic activities [56]. Failure in clearance of the residuals leads to accumulation of the neurotransmitters and ions. Excessive buildup of excitatory neurotransmitter such as glutamate is toxic to neurons and eventually causes neurodegenerative diseases [57]. High concentration of extracellular potassium ion depolarizes the membrane potential and contributes to the hyperexcitability of local neurons [58].

In the brain of HD patients, degeneration of medium spiny neurons (MSNs) in the striatum starts in the early stage of the disease before the widespread cell death in the striatum. The mechanism of the degeneration of MSNs is regulated by glutamate-mediated excitotoxicity via N-methyl-D-aspartate receptor (NMDAR) [40]. NMDAR is composed of three types of subunits (GluN1, GluN2A-D, and GluN3A-B), where NMDAR that contains GluN2A or GluN2B control synaptic dynamics [59]. Particularly, GluN2B-type NMDARs are phosphorylated by death-associated protein kinase to serve as toxic receptors [60]. Here, GluN2B-type NMDAR is highly expressed in mature MSNs of striatum, which is the reason behind the selective vulnerability of MSNs in HD [61,62]. Striatal neurons in HD show upregulated surface expression of GluN2B-type NMDAR due to the dysfunction of the huntingtin-interacting protein 14L [63]. Otherwise, gene silencing of a glutamate receptor subunit reverses HD phenotype [64]. Excitotoxicity is also studied systemically. Considering that most glutamatergic input of striatum is from the cortex, destruction of cortico-striatal pathway is a key process in presymptomatic phase of the disease [65]. Dysfunction of MSNs increases the excitation from the cortex in the form of positive feedback loop and exacerbates the excitotoxicity [66]. Interestingly, NMDAR response in the corticostriatal synapse is rescued to normal state by astrocyte-specific reduction of mHTT in BACHD mice [67]. In this context, astrocytes are recently reviewed as a player in excitotoxicity in HD and other neurodegenerative disorders [11,35,39,40,41,42,68,69,70,71,72,73,74,75,76,77].

In HD astrocytes, mRNA level of *excitatory amino acid transporter 2 (EAAT2)* is lower compared to that of the normal group [78,79]. In addition, protein level of EAAT2 is reduced throughout whole brain [34,80,81]. Low EAAT2 level is rendered as decreased glutamate uptake [82,83,84], which results in neuronal degeneration from chronic glutamate stimulation [85]. In addition, downregulation of an inwardly rectifying potassium channel, Kir4.1, in HD astrocytes results in increased extracellular K+ concentration, which subsequently increases the resting membrane potential of nearby neurons to make the neurons more excitable [86]. Interestingly, downregulation of EAAT2 and Kir4.1 in astrocytes not only induces hyperexcitability of neurons, but also induces evoked Ca^2+^ signaling in the astrocytes [87]. Increase of evoked Ca^2+^ level in astrocytes increases sodium pump activity which further increases extracellular K+ concentration [88] (Figure 1, Table 1).

### 2.2. Alteration of Oligodendrocyte Function

Oligodendrocytes are a type of glial cells in the brain and the spinal cord which produce myelin sheaths and play an important role in maintaining axonal integrity and function. Although oligodendrocytes are less explored in HD in previous studies, defective oligodendrocyte functions and deficient myelination are commonly observed in other neurodegenerative diseases [91]. Myers et al. are the first time to show an increase of oligodendrocytes in the striatum but no changes in astrocytes in postmortem HD brains [92]. Later, the other researchers identify myelin damage and breakdown in pre-symptomatic HD patients [93,94]. The full-length myelin regulatory factor (fMYRF) is self-cleaved to N-terminal myelin regulatory factor (nMYRF) which was transferred from the ER to the nucleus. In HD, binding of mHTT to nMYRF deficits normal bindings of nMYRF transcription factor in nucleus which leads to inhibition of myelination-related genes expression and oligodendrocyte dysfunction [95,96,97] (Figure 2). On the other hand, Cui et al. show that the expression of proliferator-activated receptor-gamma coactivator (PGC)-1α is significantly downregulated in HD striatal cells and tissues. In HD, mHTT interferes with promoter binding of cAMP response element binding protein (CREB) and TATA-binding protein-associated factor (TAF), which regulate the expression of PGC-1α. This mis-binding leads to the inhibition of PGC-1α expression which may cause reduction of myelin basic protein (MBP) expression and myelination deficit [98,99] (Figure 2). In parallel with these results, Xiang et al. also show the downregulation of MBP and deficient myelination in the oligodendrocytes of R6/2 transgenic mouse model of HD, and in the striatum of PGC-1α knockout mice as well [100]. Further study is necessary to verify whether PGC-1α rescues myelination in HD models in a cell-type-specific manner.

## 3. The Role of Epigenetic Modifications and Noncoding RNAs in the Pathogenesis of HD

Better understanding of epigenetic mechanisms may provide important insights, resulting in improved therapeutic approaches for treating HD [101]. In this section, we discuss the epigenetic changes and mechanisms that are associated with the pathogenesis of HD. We focus on two main epigenetic alterations that influence chromatin structure: DNA and histone modifications [102]. DNA methylation and hydroxymethylation have been involved in different neurodevelopmental and psychiatric disorders [103,104,105]. In DNA methylation, methyl groups are transferred to the cysteine 5 position of cytosine via the action of DNA methyltransferases [106]. Ng et al. propose that mHTT has a significant effect on changing the methylation of promoter regions of *octamer-binding transcription factor 1 (OCT4)*, *sex determining region Y-box 2 (SOX2)*, and *Nanog homeobox (NANOG)* as these genes are involved in neurogenesis. Therefore, inhibition of the expression level of these genes may lead to neurogenesis impairment and cognitive decline in HD [107] (Figure 3). In addition, histone modification is another major epigenetic mechanism which plays a special role in unraveling the pathogenesis of HD. CREB binding protein (CBP) interacts with several transcription factors such as specificity protein 1 (SP1), TAF, and RNA polymerase II, and acts as a co-activator or a repressor of transcription [108,109]. The CBP can also be considered as a histone acetyltransferase which acetylates histones to alter chromatin structure [110]. The mHTT interaction with CBP blocks its transcriptional co-activator function and inherent CBP histone acetyltransferase activity [111]. Therefore, CBP sequestration and depletion are accompanied by histone hypoacetylation, resulting in neuronal transcriptional dysfunction and neurotoxicity [112,113,114] (Figure 3).

Our group has found that SET domain bifurcated histone lysine methyltransferase 1 (SETDB1/ESET), a histone H3 at lysine 9 (H3K9)-specific methyltransferase, is elevated in the striatal neurons of HD patients and HD transgenic (R6/2) mice [115]. In parallel, the level of histone H3K9me3 is increased in the striatal neurons of HD patients and in HD transgenic (R6/2) mice. This study has proven that the SETDB1-H3K9me3 pathway is involved in silencing of genes in HD. Interestingly, not only SETDB1 modulates the nuclear gene transcription though heterochromatin remodeling, but it also down regulates the nucleolar gene transcription (ribosomal DNA components) by increasing methylation of upstream binding protein 1 (UBF1). SETDB1 interacts with UBF1 and trimethylates at lysine 232/235 in the nucleolus of striatal cells. As a result, trimethylated UBF1 leads to nucleolar chromatin condensation and down regulates the transcription of ribosomal DNA (rDNA) [12]. This study presents a novel epigenetic mechanism that SETDB1-UBF1 trimethylation pathway is associated the nucleolar chromatin remodeling and dysfunction of rDNA transcription in the pathogenesis of HD.

Moreover, several studies have focused on microRNAs (miRNAs) which are involved in the early differentiation, development, and function of neurons [116,117]. miR-146a is one of the major regulators of the NF-κB pathway which can also target human and mouse HTT gene [118,119]. Das et al. demonstrate that heat shock factor 1 is regulated by this miRNA, resulting suppression of mHTT aggregates in HD cells [120]. Another study confirmed that miR-214 directly targets the HTT gene which can suppress mHTT aggregation in an HD mouse striatal cell and HEK293T cell [120,121]. On the other hand, Bucha et al. showed the upregulation of miRNA-214 in HD cells could regulate mitofusin2, resulting in alteration of mitochondrial morphology [122]. Therefore, this miRNA can be considered as a critical node for therapeutic targets in HD pathogenesis.

## 4. Roles of Wild Type HTT (wtHTT) Versus mHTT in Vesicle Trafficking

Understanding the exact molecular and cellular functions of wtHTT and mHTT is crucial in further clarifying the pathogenesis of HD. wtHTT is involved in axonal transport, which is essential for neuronal synaptic activity [123]. Transport of cargo is important for neuron to work properly because of its unique morphology containing axons and dendrites. Vesicular transport is accelerated by overexpression of wtHTT [124]. There are emerging models to explain how wtHTT coordinates vesicular transport and ongoing studies to discover a more detailed mechanism of coordination and the HD pathology related to vesicle transport [125] (Figure 4).

wtHTT recruits glyceraldehyde-3-phosphate dehydrogenase (GAPDH) to transport vesicles, where vesicular GAPDH produce adenosine 5′-triphosphate (ATP) to provide energy for the transport [126]. In HD pathogenesis, GAPDH is sequestered by mHTT [127,128], where the sequestration of GAPDH is rescued by high-affinity RNA aptamers that specifically recognize mHTT [129]. In addition, HTT forms complexes of motor proteins with huntingtin-associated protein-1 and p150^Glued^ subunit of dynactin [130]. HTT-associated protein-1 binds to both kinesin-1 and vesicles to serve as an adaptor [131]. Huntingtin’s recruitment of kinesin-1 is governed by the phosphorylation of wtHTT at serine421 (Ser421), which stimulates anterograde transport [132]. Interestingly, phosphorylation of HTT (Ser421) protects against the mHTT toxicity, where the endogenous level of phosphorylated HTT (Ser421) is least in the striatum [133].

Defects in the axonal transport are associated with neurodegenerative diseases. For example, mutations in amyloid beta precursor protein obstruct motor protein activity of the hippocampal and cortical neurons in AD, and mutation in superoxide dismutase type-1 impede binding of motor proteins to neurofilaments of motor neurons in ALS [134]. In fact, fast axonal transport is commonly disrupted in polyglutamine-expansion diseases [135]. In HD, fast axonal transport is slowed down specifically in striatal neurons [136]. The impairment of the vesicular transport induces axonal degeneration, which is the early neuropathology of HD [137].

Pathogenic HTT disrupts the motility of vesicle complex, accessory proteins, and molecular motors [138], hence, the efficiency of vesicle trafficking [131,139] (Figure 4). Comparably, both HTT-depleted neurons and mHTT-expressing neurons suffer from defective axonal transport [140], where motor proteins are sequestered by mHTT [141]. Tubulin acetylation is also reduced in HD resulting in reduced binding of motor proteins to microtubules [142]. Vesicle trafficking related proteins such as HTT interacting protein 1, dynamin, and endophilin-A3 are depleted by mHTT bodies [143].

On the other hand, mHTT activates axonal c-Jun amino-terminal kinase3 via stress-signaling kinase [144], where inhibition of the c-Jun amino-terminal kinase/c-Jun partially restores striatal neurodegeneration in HD [85]. Consequently, kinesin-1 is phosphorylated at serine 176, which results in detachment of kinesin-1 and cargo from the microtubules [145].

Despite the growing evidence of mHTT and its effects on vesicle trafficking in neurons, further study is necessary to define why MSNs are much more susceptible to mHTT than other neuronal cell types. Additionally, precise cellular and molecular mechanism of mHTT oligomers versus mHTT aggregates-dependent vesicle trafficking should be determined.

In addition, wtHTT is also involved in autophagy as reviewed in [146]. wtHTT form complex with sequestosome 1 to enhance cargo recognition, where depletion of wtHTT results in empty autophagosome [147]. C-terminal domain of wtHTT has structural homology with yeast autophagy scaffold protein 11 and both proteins show similar protein–protein interaction patterns [148]. Interestingly, deletion of N-terminal domain of wtHTT in mouse suffers from DNA damage in striatum and cortex without any difference in autophagy function [149]. wtHTT is also associated with ER, where ER stress release the wtHTT to promote autophagy (reviewed in [150]). In addition, wtHTT has important role in homeostasis of presynaptic and postsynaptic terminal [151]. Loss of wtHTT lead to dysfunction of synaptic vesicle endocytosis in striatal neurons [152].

We need to provide attention to an important HD pathophysiology that the dysfunctions of central nervous system and other organs in HD are caused by mHTT accumulation as well as by the loss of functionality of wtHTT protein. Molecular simulation reveals that mHTT oligomer also sequester wtHTT [153]. Indeed, wtHTT protein expression level is inversely correlated to the age of onset [154]. In macrophage, reduced wtHTT level is associated with decreased cytokine and increased phagocytosis [155]. Research is ongoing to reveal the wtHTT function and structure further. RNA-seq of wtHTT knockout neural cell shows that wtHTT has a role in development of neurons and neurotransmission [156]. Cryo-electron microscopy structure of wtHTT confirms its role in protein–protein interaction [157]. The importance of the loss of functionality of wtHTT is associated with clinical safety of HTT gene therapy as reviewed previously [158].

## 5. Mitochondria Dysfunction in HD

A growing body of evidence show that mitochondrial dysfunctions, including membrane potential and respiratory function deficits, Ca^2+^ buffering capacity reduction, and mitochondrial number and morphology alteration, play a critical role in HD pathogenesis [159,160,161,162,163,164,165]. The mHTT aggregation reduces the mitochondrial membrane potential and increases the level of mitochondrial matrix Ca^2+^ loading that leads to decreased ATP level and enhanced reactive oxygen species (ROS) [160,166,167]. Moreover, the release of cytochrome c from dysfunctional mitochondria leads to activation of caspases 9 and 3 which are involved in apoptosis, resulting in neuronal cell death [168,169] (Figure 5). On the other hand, Yablonska et al. show mHTT binding with high affinity to translocase of mitochondrial inner membrane 23 (TIM23) complex in mitochondrial intermembrane space leads to inhibition of import of nuclear-encoded proteins through TIM23 [14]. Therefore, the mHTT–TIM23 complex interaction alters mitochondrial proteome, resulting in mitochondrial dysfunction in HD [170] (Figure 5). In addition, Guo et al. showed that valosin-containing protein (VCP) is bound to mHTT as a binding protein on the mitochondria. Mitochondria-accumulated VCP works as a mitophagy adaptor to bind to the autophagosome component, microtubule-associated proteins 1A/1B light chain 3B (LC3), leading to enhanced mitophagy, reduced mitochondrial mass, and ultimately, neuronal cell death [28,171] (Figure 5). Moreover, the previous studies demonstrated that down regulation of wtHTT is related to mitochondria dysfunction by inability of the mitochondria to generate ATP [172] and diminished purines and inosine monophosphate [6].

## 6. Therapeutic Approaches for Huntington’s Disease

Despite remarkable efforts to overcome the symptoms of HD, effective therapeutic targets are still very limited in HD. Furthermore, no standard treatment has been established for HD. HD transgenic mouse models have been used for translational study with many candidate drugs before conducting clinical trials with patients, but the efficacy of most drugs is lower than expected [173]. Accordingly, the benefits of translating the therapeutic efficacy from the HD transgenic mouse models to human patients are not clear.

Gene editing method and strategy have been attempted for treating various genetic disorders including HD. Clustered regularly interspaced short palindromic repeats and (CRIPSR) and CRISPR-associated genes (CRISPR/Cas9) has been applied haplotype-specifically to common promoter-local single-nucleotide polymorphisms (SNPs) for the selective deletion of *mHTT* [174,175,176,177]. Otherwise, Zinc finger proteins containing the Kruppel associated box (KRAB-ZFPs), short hairpin RNA (shRNA), small interfering RNA (siRNA), and miRNA have been examined to impair transcription of mHTT [178,179,180,181,182,183,184,185]. To use siRNA or antisense oligonucleotide to knock-down mRNA of mutant Hunting requires repetitive administration. shRNA treatment lasts relatively longer than siRNA treatment; however, the dosage control of shRNA treatment is limited [186]. miRNA therapy suffers from off-target effect in general, which is recently overcome using in silico analysis [187]. Glutamine repeat-binding [188] and deletion [189] on *mHTT* gene are also effective in HTT lowering and alleviate HD phenotypes. For further information, gene targeting approaches are reviewed in [190,191,192,193,194,195,196,197,198,199,200,201].

Among many small compounds, epigenetic modulators have been used for rescuing transcriptional dysfunction in HD. For example, phenylbutyrate, sodium butyrate, histone deacetylases inhibitor (HDACi) 4b and LBH589, Tubastatin A, and CKD-504 hinder histone deacetylase increase the acetylation of H3K9, and improve neuropathology, behaviors, and survival of HD transgenic mice [202,203,204,205,206,207,208]. Importantly, epigenetic compounds exhibit transgenerational effect in HD animal models [205]. Mithramycin, a DNA binding drug, inhibits expression of histone methyltransferase, reduces H3K9me3 level and heterochromatin condensation, and ameliorates symptoms of HD [209]. In order to improve the efficacy of epigenetic compounds, further efforts to reduce the side effect of these drugs need to be made.

In addition to HD genetic and epigenetic targets, pathologic phenotypes including HTT fragmentation and aggregates, transcriptional dysfunction, oxidative stress, apoptosis, autophagy dysfunction, and excitotoxicity appear to be reasonable drug targets [101] (Table 2). The most effective therapeutic strategy in HD is to target the inhibition of aggregation or fragmentation of mHTT, because mHTT is directly responsible for the pathogenesis of HD. Cystamine, Congo red, Chrysamine G, direct fast yellow, and trehalose are drugs that bind to polyglutamine or block oligomerization, and consequently inhibit the aggregation of mHTT. Congo red, an organic compound and diazo dye, binds to β–sheets of protein structure of mHTT and prevents polyglutamate oligomerization. Since Congo red cannot cross the blood–brain barrier, compounds with similar structure of Congo red are discovered as potential drugs. Chrysamine G and direct fast yellow are found to effectively inhibit mHTT aggregation [210]. Saccharides including trehalose also bind directly to the polyglutamate region of mHTT to suppress the mHTT aggregation effectively [211]. Both antibodies which binds polyproline domain of mHTT and DnaJ heat shock protein family member B6 also reduce mHTT aggregation [212]. Otherwise, insulin, exendin-4, GM1, RCAN1-IL, and SGK block mHTT aggregation by increasing mHTT phosphorylation and modifying mHTT toxicity through post-translational modification. Increasing the phosphorylation of mHTT enhances solubility and decreases aggregation. Surprisingly, phosphorylation of mHTT on Ser 421 is known to be neuroprotective [213,214]. Formation of mHTT aggregate is exacerbated by transglutaminase, which cross-links mHTT. Inhibition of transglutaminase with cystamine reduces abnormal behavior, extends lifespan, and prevents weight loss of HD transgenic (R6/2) mice. In addition, cystamine injected HD mice have higher expression level of Dnajb1, which catalyze ATP hydrolysis. In addition to the mHTT aggregation, fragmentation of mHTT has been a plausible therapeutic target for HD because the pathology of HD is exacerbated by mHTT fragments including polyglutamate region [215]. Minocycline and Z-VAD-FMK inhibit caspases to prevent the proteolysis of mHTT and improve neuropathology of HD transgenic mice [216].

Interestingly, Rieux et al. (2020) tested whether a parabiosis therapy, an in vivo blood transfusion via surgical linking of two bodies, can reduce mHTT propagation and pathology in HD transgenic mice (zQ175 mice) [217]. It is concluded that blood transfusion improves mitochondrial activity in peripheral organs and ameliorates neuropathology in MSNs of striatum. This study indicates that healthy blood can diminish the pathogenicity of circulating mHTT. If the concentration of mHTT exceeds a certain concentration in the body, it is likely to cause a disease onset of HD systemically. In this paradigm, reducing the concentration of mHTT by removing circulating mHTT with blood transfusion can be another treatment. However, application of the parabiosis therapy for HD may need further verification in regard to unexpected adaptive immune reactions in vivo.

Mitochondrial dysfunction is also one of the therapeutic targets in HD. Creatine is applied to restore mitochondrial dysfunction as it deactivates mitochondrial permeability transition [218]. Coenzyme Q10 promotes electron transport chain activity, which in turn improves mitochondrial respiration [218,219,220]. Both creatine and coenzyme Q10 have been used as beneficial compounds in HD and progressed up to Phase II clinical trials. Mitochondrial dysfunction induces oxidative stress, which can be managed by antioxidants (reviewed in [221]). PGC-1α is associated with transcriptional regulation of mitochondria-related genes and is also the target of HD therapy (reviewed in [222]). rhIGF-1 increases glucose uptake and regulates energy metabolism in striatal neurons, and its therapeutic effect has been tested in HD transgenic mouse models (R6/2 and YAC128) [223]. Autophagy is also involved in the clearing and recycling of mHTT in MSNs and its function is impaired in HD [224]. Niclosamide reduces mHTT by increasing autophagy activity [225]. It seems likely that niclosamide is therapeutically more effective in increasing lysosomal degradation of ubiquitinated molecules including ubiquitinated mHTT rather than activating proteasomal activity [226]. Apoptotic cell death of MSNs has been a therapeutic target in HD [227]. MAP4343, 17EE2, and isoquercitrin are known to control stress responses and reduce apoptosis of MSNs in HD [228]. Z-VAD-FMK, Z-DEVD-FMK, Z-LEHD-FMK, PG3d, and lithium chloride are well-known apoptosis inhibitors and used to treat HD animal models (*C. elegans* and Rat) [216,226,227,229,230]. Laquinimod increase the brain-derived neurotrophic factor level in striatum of R6/2 mouse model and has the neuroprotective effect [231].

Continuous stimulation by excitatory or inhibitory neurotransmitter can damage MSNs in HD. Notably, controlling glutamate-induced neurotoxicity is one of many therapeutic strategies for treating HD and other neurodegenerative disorders (reviewed in [232]). Silencing of a glutamate receptor subunit could reverse HD phenotype [64]. Activation of NMDA receptors and cation channels elevates intracellular Ca^2+^ flux, impairs mitochondria function, and triggers neuronal cell death pathways. Memantine acts as an inhibitor of NMDA receptor, draws attention in HD therapy, and its clinical trials are on-going at phase 2 and phase 4, respectively. Otherwise, necrostatin-1, an inhibitor of receptor-interacting serine/threonine-protein kinase 1 (RIPK1) and necrosis, shows positive effects for delaying the onset and improving motor behaviors while the survival extension is not improved in HD transgenic (R6/2) mice [77,233,234].

## 7. Conclusions

Since the mutation of *HTT* gene at exon 1 with glutamine repeats was identified as the cause of HD in 1993 [2], many studies have shown that mHTT proteins directly cause the neuropathogenesis of HD. wtHTT plays an important role in vesicular transport, which is an essential cellular event in MSNs, whereas mHTT disrupts vesicle transport by sequestrating motor proteins. Understanding of exact mechanisms on the mHTT-induced selective neuronal damage in the neostriatum is pivotal to develop beneficial therapeutic targets or strategies to ameliorate the neurodegeneration in HD. In this context, further investigations about effective clearance or detoxification of mHTT remain to be performed.

The brain is a multicellular organ. Accordingly, it is possible that mHTT-induced cellular dysfunctions are varied and differentially modulated in specific brain regions and cell-type specific manner. In terms of autonomous versus non-cell autonomous neuronal damage, it is also critical to determine which brain cell-types (e.g., excitatory neurons, inhibitory neurons, astrocytes, and oligodendrocytes) are vulnerable to mHTT and contribute to the pathogenesis of HD [11,70]. Importantly, gliosis, production of new astrocytes, microglia, and oligodendrocytes, is a prominent pathology in HD as well as in other neurodegenerative disorders (reviewed in [74,77]). Therefore, it is necessary to define how mHTT affects the fate of neuron and glia, and whether therapeutic targets can selectively modulate and rescue cell-type specific functions in HD.

The scope of this review is to briefly introduce the previous and recent studies about mechanisms of HD pathologies and therapeutic strategies and that our review has limitation in the scope. Just in case, for the readers need further information, we recommend the previous reviews dealt with the specific aspects of striatal vulnerability [248], white matter phenotype [249], cerebellar dysfunction [250], progression of cell type-specific phenotype [251], microglial activation [252], synapse [253], intracellular transmission of mHTT [254], protein–protein interactions [255], biochemical alterations and HTT dynamics [256,257], posttranslational modifications [258], proteostasis [259], autophagy [260,261], redox homeostasis [262], metabolism [263,264], *HTT* mRNA [265,266], Ca^2+^ and dopamine signaling [61], inflammation [267], in vitro modelling of HD [268,269], striatal neurogenesis [270], stem cell treatment [271,272,273,274,275,276,277,278,279], electric stimulation therapy [280], network connectivity in presymptomatic HD brain [281], non-motor symptoms [282], gut microbiome [283], human immunodeficiency virus [284], diagnosis [285,286], clinical progression [287], treatment for the symptoms [288], physical therapy [289], psychological interventions [290,291], and management of agitation [292]. Collectively, the previous studies have potential to reveal spatiotemporal and cell-type specific mechanism of HD pathology. The future challenges in HD research are brought by the complexity of the pathology from biochemical level [293,294,295,296,297,298,299,300,301,302,303] to system level [304,305,306,307,308]. Accordingly, the ultimate mechanisms of HD pathology can be further scrutinized by state-of-the art research methods such as multi-omics approach combining transcriptome, proteome, and interactome [309], big data analysis with machine learning [310], and meta-analysis combining the publicly available data [311]. On the other hand, the potential HD therapeutics should specifically modulate the function of the striatal neurons while they prevent the adverse behavior of glial cells. High-throughput in silico and in vitro screening of chemical libraries [312,313,314,315,316] are expected to expedite the designing of beneficial compounds for HD.

Previous studies indicate that epigenetic cellular events have been emerged as potential therapeutic targets in HD [12,109]. The reversible characters of epigenetic modifications during the pathogenesis of HD are reasonable therapeutic targets. It is highly expected that we can prevent neuronal damage more efficiently by balancing the epigenetic disequilibrium in HD before the pathogenesis becomes irreversible and degenerative under HD stress condition. In this regard, future therapeutic strategies and agents to treat HD should consider appropriate epigenetic targets and cell-type specificity. On the other hand, identification of blood cell-derived epigenetic markers that can mimic the brain molecular pathology, will facilitate the advanced diagnosis and treatment of HD. Taken together, development of cell-type specific epigenetic therapeutic targets will pave a way to slow down the onset and progress of HD.

## Figures and Tables

**Figure 1 ijms-22-12499-f001:**
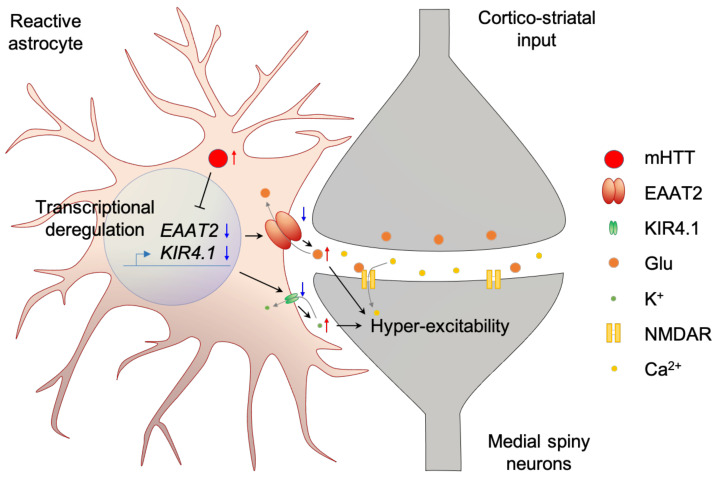
Dysfunction of cortico-striatal tripartite synapse in HD. mHTT aggregation deregulates transcription of EAAT2 and Kir4.1 in astrocytes. Deregulation of EAAT2 leads to low expression level of astrocyte glutamate transporter and impairment of the glutamate uptake by astrocytes in the tripartite synapse. As a result, excess glutamate in synapse induces hyper-excitability of the post-synaptic neuron (medial spiny neuron in cortico-striatal synapse). Deregulation of Kir4.1 leads to impaired potassium buffering of astrocytes. As a result, elevated potassium ion concentration increases the membrane potential of neurons, where high membrane potential induces hyper-excitability. Prolonged hyper-excitability is rendered as cellular toxicity.

**Figure 2 ijms-22-12499-f002:**
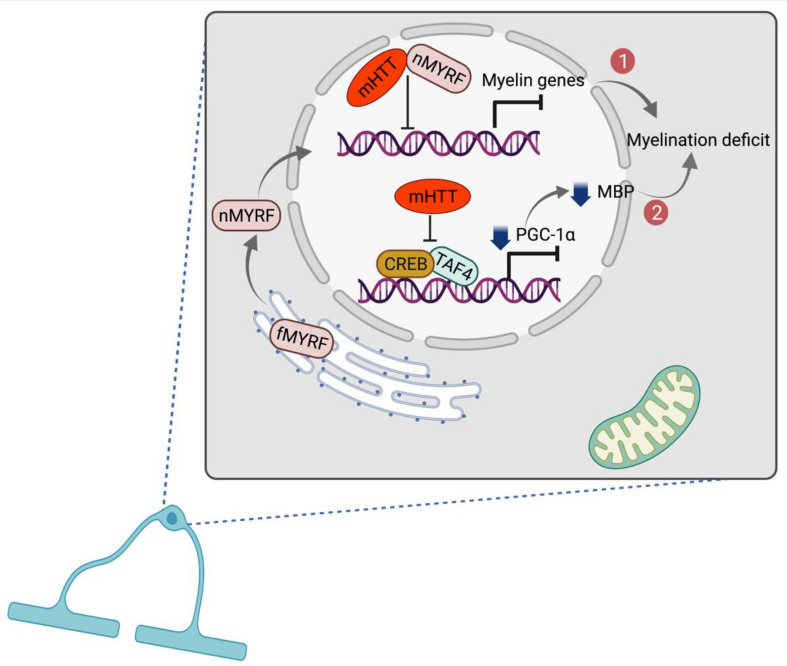
Oligodendrocyte dysfunction in HD. In the first pathway, full-length MYRF is self-cleaved to nMYRF which detaches from ER and is translocated to the nucleus to regulate the expression of myelin related genes. In HD, N-terminal mHTT binds nMYRF causing abnormal binding of nMYRF and deficit myelin genes expression. Second pathway shows inhibition of PGC-1α expression by interference of mHTT in co-binding of CREB and TAF4, leading to reduced activity in the cholesterol biosynthesis pathway and myelination deficit. Created with BioRender.com.

**Figure 3 ijms-22-12499-f003:**
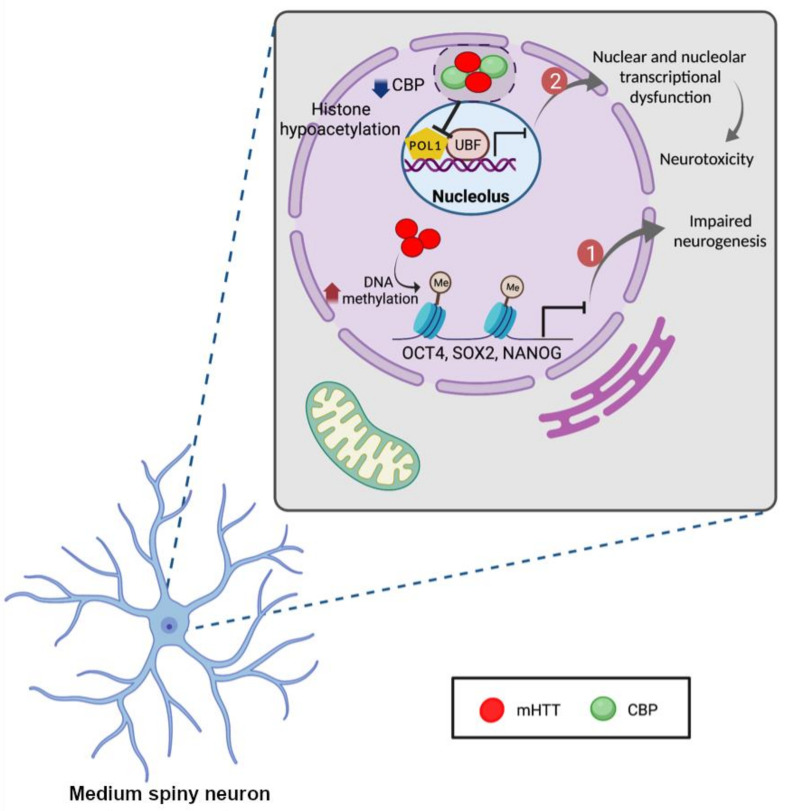
Epigenetic modifications associated with HD. The promoter regions of neurogenesis-related genes, *OCT4*, *SOX2*, and *NANOG*, are methylated in cells expressing mHTT which can lead to impaired neurogenesis. On the other hand, mHTT sequestrates CBP in nuclear inclusions which causes the hypermethylation and hypoacetylation of histone proteins and CBP depletion. Depletion of CBP from the nucleus of cells leads to histone hypoacetylation, nuclear and nucleolar transcriptional dysfunction and increase in neurotoxicity. Created with BioRender.com.

**Figure 4 ijms-22-12499-f004:**
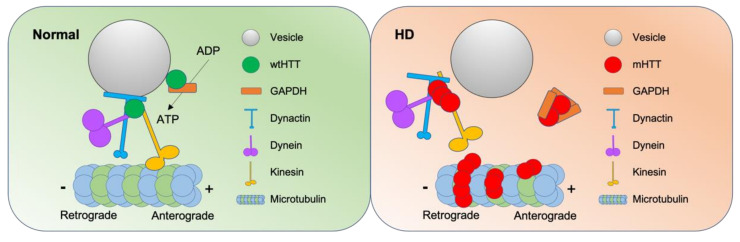
Role of HTT in normal vesicular transport and role of mHTT in disturbed vesicular transport in HD. In normal conditions, HTT participates in motor protein complex with dynactin, dynein, and kinesin. In addition, HTT recruits GAPDH to vesicles to supply energy, ATP to motor proteins. In HD, polyglutamine expansions of the mHTT sequester GAPDH and motor proteins. Microtubules are acetylated by mHTT to hinder binding of motor proteins on the microtubules.

**Figure 5 ijms-22-12499-f005:**
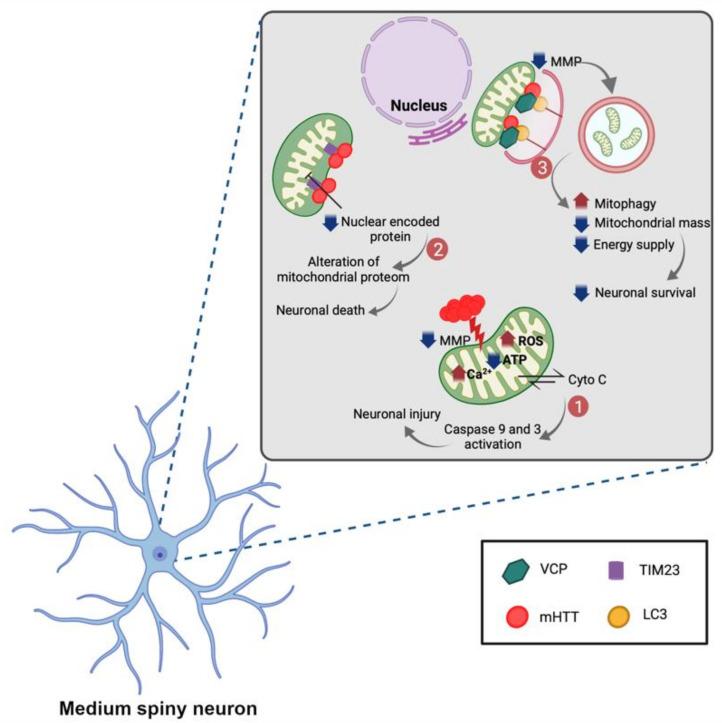
Mitochondria dysfunction associated with HD. mHTT aggregation disrupts the mitochondrial membrane potential and increases excitotoxin-induced Ca2+ influx which leads to decreased ATP generation and increased ROS. Mitochondria dysfunction results in release of cytochrome C from the mitochondria which triggers the activation of apoptotic cascade via caspases 9 and 3, and neuronal injury. On the other hand, in HD, mHTT binds with high-affinity to TIM23 in mitochondrial intermembrane space, causing diminished levels of nuclear-encoded proteins imported through TIM23 and subsequently, neuronal death. Finally, VCP is selectively translocated to the mitochondria, where it is bound to mHTT and LC3 to enhance mitophagosome production, and reduce mitochondrial mass and energy supply, causing neuronal cell death. Created with BioRender.com.

**Table 1 ijms-22-12499-t001:** Non-cell autonomous cell death pathway in HD related to astrocytes.

HD Pathology	Specimen	Brain Region/Cell Type	Experimental Method	Reference
Less *EAAT2* mRNA	Postmortem	Cingulate cortex	RNA sequencing	[78]
		Neostriatum	In situ hybridization	[79]
Less EAAT2 protein	Postmortem	Striatum	Immunohistochemistry	[34]
		Striatum, cortex	Western blot	[80]
	Mouse; R6/2	Striatum, cortex, hippocampus, midbrain	Quantitative proteomics	[81]
Less glutamate uptake	Postmortem	Prefrontal cortex	Glutamate uptake assay	[84]
	Cell; astrocyte	differentiated from Q77 monkey iPSC	Glutamate uptake assay	[82]
	Mouse; Q175	Single corticostriatal synapse	Imaging assay with glutamate sensor	[83]
Less *Kir4.1* mRNA	Postmortem	cingulate cortex	RNA sequencing	[78]
Less Kir4.1 protein Higher extracellular K^+^More excitable	Mouse; R6/2	Striatal MSN and astrocyte	qPCR, IHC, Western blot, Virus microinjection, Electrophysiology	[86]
Altered Ca^2+^ signal	Mouse; R6/2	Striatal astrocyte	Virus microinjectionElectrophysiology	[87]
More excitotoxicity	Cell; neuron and astrocyte	Co-culture of HD neurons and astrocytes from human iPSC	Cell count after glutamate exposure	[89]
		Co-culture of wild type neurons with mHTT infected glia from rat primary culture		[90]

**Table 2 ijms-22-12499-t002:** Therapeutic targets for HD.

Target	Strategy	Mode of Action	Disease Model	Clinical Trial & NCTno.	References
*mHTT* gene	CRISPR/Cas9	Excise *mHTT* DNA selectively	Cell iPS		[174]
			Mouse BacHD		[175]
			Mouse HD140Q		[176]
			Mouse R6/2		[177]
	KRAB-ZFPs	Inhibition of translation or transcriptdegradation	Mouse R6/1,2		[178,179]
	shRNA		Mouse R6/2		[180]
			Mouse N171-82Q		[181,182]
	siRNA		Mouse *HTT* injected		[183]
			Mouse Hdh-150Q		[184]
	miRNA		Mouse HD140Q	Phase I/II [235], NCT04120493	[185]
	Antisense nucleotide	Bind to *HTT* mRNA	Mouse BACHD	Phase II, NCT02519036	[236,237]
Transcriptional dysregulation	Phenylbutyrate	Inactivate histone deacetylase	Mouse N171-82Q	Phase II, NCT00212316	[202]
	Sodium butyrate		Mouse R6/2		[203]
	HDACi 4b		Mouse N171–82Q		[204,205,206]
	HDACi LBH589		Transgenic Rodent HD Models		[207]
	Tubastatin A		Cell primary neuron		[208]
	CKD-504			Phase I, NCT03713892	
	Mithramycin	Increase H3K9	Mouse R6/2		[209]
mHTT aggregation	Cystamine	Suppress mHTT crosslinking	Mouse R6/2		[238,239]
	Congo red	Bind and inhibit polyglutamineoligomerization	Mouse R6/2		[240]
	ChrysamineG, Direct fast yellow				[210]
	Trehalose				[211]
mHTT fragmentation	Minocycline	Inhibit caspase	Mouse R6/2	Phase III, NCT00277355	[241]
	Z-VAD-FMK		Cell X57		[216]
mHTT lowering	Blood transfusion	Remove circulating mHTT	Mouse zQ175		[217]
mHTT post-modification	Insulin, exendin-4	Increase mHTT Phosphorylation	Cell SH-SY5Y		[242]
	GM1		Mouse YAC128		[243]
	RCAN1-1L		Cell ST14A		[244]
	SGK		Cell primary neuron		[245]
Transactivation	KD3010	Increased PPARδ transactivation	Mouse pCAGGS-loxP-STOP-loxP		[246]
Mitochondrial dysfunction	Creatine	Inactivate mitochondrial permeability transition	Mouse R6/2	Phase II, NCT00026988	[218]
	Coenzyme Q10	Enhance electron transport		Phase II, NCT00920699	[219,220]
	PGC-1α	Upregulate mitochondrial gene			[222]
Metabolism	rhIGF-1	increase glucose uptake	Mouse YAC128 Mouse R6/2		[223]
Autophagy	Niclosamide	Inhibit mTOR	Cell HEK293, N2a		[225]
	MAP4343, 17βE2, Isoquercitrin	Regulate stress response	*C.**elegans* HD mutants		[228]
Apoptosis	Z-VAD-FMK, Z-DEVD-FMK, Z-LEHD-FMK	Inhibit caspase	Cell primary neuron		[227]
	PG3d		Cell COS-7		[229]
	Lithium chloride		Rat QA injected		[230]
Excitotoxicity	Memantine	Inhibit NMDA receptor		Phase IV, NCT00652457; Phase II, NCT00652457	
	Necrostatin-1	Inhibit RIP1 kinase	Mouse R6/2		[233,247]

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
