# Peer review of "Non-Cell Autonomous and Epigenetic Mechanisms of Huntington’s Disease"

_ijms, 2021, doi:10.3390/ijms222212499_

Round 1

Reviewer 1 Report

Manuscript by Kim and Yousefian-Jazi et al. refers to important aspects of pathogenesis of Huntington’s disease. Almost 30 years from identification of gene which is mutated in HD, still many aspects of neurodegeneration are not well understood. Indeed, additional mechanisms are considered to contribute to overall pathogenesis.

Some sentences need to be improved to present the information more clearly. From reading of some parts I had an impression that manuscript was not very carefully checked. More precision and consistency is required. I suggest all Authors to re-read the manuscript for additional improvement. The examples are:

  • Lines 34-35, “The mHTT interactomes involve transcriptional alterations, intracellular transport, (…)”. I am not sure about the intended meaning here, is it about interactions of mHTT which lead to transcriptional alterations? And that mHTT-interaction factors are also involved in intracellular transport?
  • Lines 37-38, “Epigenetic modifications which can alter HD transcriptome (…). More precisely, I would rather say about epigenetic modifications which are altered in HD cells and affect the level of transcripts/gene expression.
  • Lines from 66 and from 90, information about GAPDH is given in two separated paragraphs.
  • Line 251: “mHTT is directly and indirectly responsible for the pathogenesis of HD” (as well as line 332). I am not sure about the meaning here and I think it is confusing. mHTT as a product of mutated gene leads directly to disruptions in molecular pathways, what leads to further disruptions etc., but I would rather refer that mHTT is directly responsible for HD pathogenesis.
  • Chapter 6. Information about therapeutic approaches could be better organized, starting from description of approaches which are design to affect expression of mutant gene.
  • Lines from 286: Gene targeting approaches are described very shortly, as compared to other. Using antisense oligonucleotides is not mentioned (only in Table). shRNA, siRNA and miRNAs usually do not impair transcription but cause transcript degradation or inhibition of translation. As there are numerous studies, I would recommend to refer to some recent reviews in this topic.
  • Concerning mHTT gene-targeting approaches there are numerous studies not mentioned, including additional clinical trials not listed in Table. I do not find this information crucial in this review. The author should decide whether include more complete information or only shortly mention these issues without details.

For 177 references cited, about 60 is from 2016-2021. I think it would be worth to include more recent studies, leaving only the most important older references.

Some minor comments:

  • Line 38, “therapeutics” should be rather “therapeutic targets”
  • Line 47, “can occur” instead of “can be occurred”
  • For consistency with mHTT it would be better to write: wtHTT
  • Chapter 3.: mHTT is named here polyglutamine-expanded HTT

Author Response

We greatly thank Reviewer A for the thoughtful and constructive comments. We addressed all of her/his concerns in the revised manuscript.

Manuscript by Kim and Yousefian-Jazi et al. refers to important aspects of pathogenesis of Huntington’s disease. Almost 30 years from identification of gene which is mutated in HD, still many aspects of neurodegeneration are not well understood. Indeed, additional mechanisms are considered to contribute to overall pathogenesis.

Some sentences need to be improved to present the information more clearly. From reading of some parts I had an impression that manuscript was not very carefully checked. More precision and consistency is required. I suggest all Authors to re-read the manuscript for additional improvement. The examples are:

  1. Lines 34-35, “The mHTT interactomes involve transcriptional alterations, intracellular transport, (…)”. I am not sure about the intended meaning here, is it about interactions of mHTT which lead to transcriptional alterations? And that mHTT-interaction factors are also involved in intracellular transport?

Response: This sentence is about the involvement of mHTT in different aspect of HD pathogenesis which is reviewed in this review manuscript such as transcriptional alterations, intracellular transport, excitotoxicity, collapse of the protein degradation mechanisms, mitochondrial dysfunction, and disorders of myelin.

We also changed the sentence to avoid confusion as follows (lines 43-47):

“The mHTT is involved in transcriptional alterations, disruption of intracellular transport, excitotoxicity, collapse of the protein degradation mechanisms, mitochondrial dysfunc-tion, and disorders of myelin, that make neurons more susceptible to generic stresses, eventually leading to neuronal death [11-14].”

  1. Lines 37-38, “Epigenetic modifications which can alter HD transcriptome (…). More precisely, I would rather say about epigenetic modifications which are altered in HD cells and affect the level of transcripts/gene expression.

Response: The sentence was modified as identified below (lines 54-57):

“The epigenetic modifications are closely associated with the pathogenesis of HD [17,18]. mHTT sequesters specific transcription factors and impairs their function, resulting in disruption of their target genes transcription [19-21]. mHTT also regulates transcription by facilitating transcriptional factor interaction with protein complexes [22].”

  1. Lines from 66 and from 90, information about GAPDH is given in two separated paragraphs.

Response: In response to the reviewer’s suggestion, we reorganized the information about GAPDH into the following sentences (lines 240-244).

“wtHTT recruits glyceraldehyde-3-phosphate dehydrogenase (GAPDH) to transport vesicles, where vesicular GAPDH produce adenosine 5’-triphosphate (ATP) to provide energy for the transport [126]. In HD pathogenesis, GAPDH is sequestered by mHTT [127,128], where the sequestration of GAPDH is rescued by high-affinity RNA aptamers that specifically recognize mHTT [129].”

  1. Line 251: “mHTT is directly and indirectly responsible for the pathogenesis of HD” (as well as line 332). I am not sure about the meaning here and I think it is confusing. mHTT as a product of mutated gene leads directly to disruptions in molecular pathways, what leads to further disruptions etc., but I would rather refer that mHTT is directly responsible for HD pathogenesis.

Response: In response to the reviewers’ suggestion, we rewrote the first sentence as shown below (lines 369-371 and lines 440-442:

“The most effective therapeutic strategy in HD is to target the inhibition of aggregation or fragmentation of mHTT, because mHTT is directly responsible for the pathogenesis of HD.”

“Since the mutation of HTT gene at exon 1 with glutamine repeats was identified as the cause of HD in 1993 [2], many studies have shown that mHTT proteins directly cause the neuropathogenesis of HD.”

  1. Chapter 6. Information about therapeutic approaches could be better organized, starting from description of approaches which are design to affect expression of mutant gene.

Response: As the reviewer suggested, we reorganized the Chapter 6 with subchapters into the following paragraphs (lines 336-355):

“Despite remarkable efforts to overcome the symptoms of HD, effective therapeutic targets are still very limited in HD. (…) For further information, gene targeting approaches are reviewed in [191-202].”

  1. Lines from 286: Gene targeting approaches are described very shortly, as compared to other. Using antisense oligonucleotides is not mentioned (only in Table). shRNA, siRNA and miRNAs usually do not impair transcription but cause transcript degradation or inhibition of translation. As there are numerous studies, I would recommend to refer to some recent reviews in this topic.

Response: As the reviewer suggested, we added more information regarding gene targeting approaches as follows (lines 342-355):

“Gene editing method and strategy have been attempted for treating various genetic disorders including HD. Clustered regularly interspaced short palindromic repeats and (CRIPSR) and CRISPR-associated genes (CRISPR/Cas9) has been applied haplo-type-specifically to common promoter-local single-nucleotide polymorphisms (SNPs) of mHTT for the selective deletion of mHTT [175-178]. Otherwise, Zinc finger proteins con-taining the Kruppel associated box (KRAB-ZFPs), short hairpin RNA (shRNA), small in-terfering RNA (siRNA), and miRNA have been examined to impair transcription of mHTT [179-186]. To use siRNA or antisense oligonucleotide to knock-down mRNA of mutant Hunting requires repetitive administration. shRNA treatment lasts relatively longer than siRNA treatment, however, the dosage control of shRNA treatment is limited [187]. miRNA therapy suffers from off-target effect in general, which is recently overcome using in silico analysis [188]. glutamine repeat-binding [189] and deletion [190] on mHTT gene are also effective in HTT lowering and alleviate HD phenotypes. For further infor-mation, gene targeting approaches are reviewed in [191-202].”

  1. Concerning mHTT gene-targeting approaches there are numerous studies not mentioned, including additional clinical trials not listed in Table. I do not find this information crucial in this review. The author should decide whether include more complete information or only shortly mention these issues without details.

Response: Thank you for the kind advice. As you mentioned, numerous studies and even clinical trials are underway targeting the mHTT gene (miRNA clinical trial added in the table 2). We would like to briefly introduce the text without adding details and refer to recent reviews in this topic as we responded in 6.

  1. For 177 references cited, about 60 is from 2016-2021. I think it would be worth to include more recent studies, leaving only the most important older references.

Response: As the reviewer suggested, we replaced some references with more recent references and currently we have 181 references published from 2016-2021 among 318 references.

Some minor comments:

  1. Line 38, “therapeutics” should be rather “therapeutic targets”

Response: Corrected.

  1. Line 47, “can occur” instead of “can be occurred”

Response: Corrected.

  1. For consistency with mHTT it would be better to write: wtHTT

Response: Corrected.

  1. Chapter 3.: mHTT is named here polyglutamine-expanded HTT

Response: As the reviewer suggested, polyglutamine-expanded HTT changed to mHTT through the manuscript for consistency.

Reviewer 2 Report

06 October 2021

Review on the manuscript titled “Non-cell autonomous and epigenetic mechanisms of Huntington’s Disease” by Kim C et al., submitted to International Journal of Molecular Sciences (IJMS).

Manuscript ID: ijms-1430090

Dear Authors,

Huntington's disease (HD) is one of neurodegenerative diseases characterized by an expansion of CAG trinucleotide repeat Huntingtin (HTT) gene and mutant HTT protein neurotoxicity. The authors review the role of HTT, describe how mHTT affects neurons, epigenetic modifications, and transcription, and define non-cell autonomous pathway, presenting possible therapeutic targets.

Please reconsider the following parts:

  1. A graphic abstract summarizing the manuscript is highly recommended.
  2. Page 2, Keywords: Please list more keywords up to ten.
  3. Pages 1-2, Introduction: Please present short descriptions of epidemiology, symptomatology, pathology, biomarkers,  current treatment, and current research of HD, such as inflammation, ballooning cell death transcriptional repression-induced atypical cell death of neuron (TRIAD), among others. 
  4. Page 6, The Section 5: “Non-cell autonomous pathway”; Please define the term and present a short introduction before the subsection in the beginning of the section.
  5. Page 7, Conclusion: Please present conclusion with future perspective, including weaknesses or limitation in the present review, potentials, the ultimate goal, research or knowledge needed to achieve, the biggest challenge in this goal, and future research direction, among others.

The manuscript contains five figures, two tables, and 177 references. The manuscript carries important value exploring the mechanism of mHTT neurotoxicity and potential intervention in HD. I recommend this manuscript for publication after minor revision.

Author Response

Thanks for the valuable comments. We went through each comment carefully, revised the manuscript and responded point by point.

Huntington's disease (HD) is one of neurodegenerative diseases characterized by an expansion of CAG trinucleotide repeat Huntingtin (HTT) gene and mutant HTT protein neurotoxicity. The authors review the role of HTT, describe how mHTT affects neurons, epigenetic modifications, and transcription, and define non-cell autonomous pathway, presenting possible therapeutic targets.

Please reconsider the following parts:

  1. A graphic abstractsummarizing the manuscript is highly recommended.

Response: Thanks for your suggestion, we consider each figure in each section as a graphic abstract for the section. Then, it would be too complicated and useless if we add a graphic abstract combining all those schemes.

  1. Page 2, Keywords:Please list more keywords up to ten.

Response: Some more keywords were added:

“Huntington's disease; non-cell autonomous pathway; astrocyte; oligodendrocyte; epigenetics; mitochondria dysfunction; vesicle trafficking; therapeutic targets”

  1. Pages 1-2, Introduction:Please present short descriptions of epidemiology, symptomatology, pathology, biomarkers, current treatment, and current research of HD, such as inflammation, ballooning cell death transcriptional repression-induced atypical cell death of neuron (TRIAD), among others. 

Response: On the suggestion of the Reviewer B, we rewrote the first and second paragraph of introduction to add the description of mentioned terms (lines 28-53).

“Huntington’s disease (HD) is a fatal progressive neurodegenerative disorder with a mid-life onset which ranges from infancy to the ninth decade. HD is occurred in 5 – 10 cases per 100,000 persons worldwide, and characterized by chorea, emotional distress, and progressive cognitive decline [1]. Generally, in HD patients, there are more than 38 repeats of trinucleotide glutamine within the Huntingtin (HTT) gene with an inverse rela-tionship between the number of glutamine repeats and the age of onset, indicating that the high number of glutamine repeats cause the earlier phenotype of HD symptoms [2-4]. HTT has a crucial role in embryonic development by showing the death of embryos of huntingtin homozygous knockout mice by day 7.5 [5]. On the other hand, HTT plays an important role in cardiomyocytes cellular energy and nucleotides metabolism [6].

Currently, HD is considered as a multi-systemic neurodegenerative disease due to skeletal muscle and heart function disturbances with energy metabolism and mitochon-drial alterations beyond the brain dysfunction [7]. Although most of studies show intra-cellular inclusions formed by mutant HTT (mHTT) protein in selected regions of the brain such as the striatum and cortex of HD brain [8,9], other studies find the expression and function of mHTT in skeletal muscle and heart as well [10]. The mHTT is involved in transcriptional alterations, disruption of intracellular transport, excitotoxicity, collapse of the protein degradation mechanisms, mitochondrial dysfunction, and disorders of mye-lin, that make neurons more susceptible to generic stresses, eventually leading to neuronal death [11-14]. Yamanishi et al. reported a novel cell death pathway known as transcrip-tional repression-induced atypical cell death of neurons (TRIAD) in the HD pathology, exhibiting that enlargement of endoplasmic reticulum (ER) contributes to neuronal cell damage without alteration of mitochondria and nuclei structure [15]. Another study has shown that inflammation may play a role in cardiac dysfunction in HD by overexpression of the inflammatory cytokine, tumor necrosis factor-α, in cardiomyocytes of R6/2 HD mice [16].”

  1. Page 6, The Section 5:“Non-cell autonomous pathway”; Please define the term and present a short introduction before the subsection in the beginning of the section.

Response: As the reviewer suggested, we defined the term and added introduction in the beginning of the section as shown below (lines 74-94):

“While the primary causes of neuronal damage and cell death are abnormalities within the damaged neuron itself, other neighboring cells such as glial cells also contrib-ute to the neuronal death [37,38]. In this regard, non-cell autonomous pathway is defined in this review as the mechanism of the neuronal damage caused by non-neuronal cells. Recent studies have proven the non-cell autonomous pathway in that reactive astrocytes lead to neuronal damage and neurodegeneration in Alzheimer’s disease (AD), amyo-trophic lateral sclerosis (ALS), HD, and Parkinson’s disease (PD) [39-42]. Indeed, striatal or cortical neuron-specific expression of mHTT alone is not sufficient to fully induce pathological phenotypes of HD [43]. Otherwise, astrocyte-specific expression of mHTT does not induce neurodegeneration, but it shows neurological symptoms [44]. In the HD model where mHTT is selectively expressed in neurons, selective expression of a mHTT-aggregation suppressor in astrocytes expands the lifespan [45]. On the other hand, white matter is degenerated and the oligodendrocyte differentiation is defective in HD [46]. mHTT-expressing microglia are hyper-reactive to inflammatory stimuli to cause synaptic dysfunction in dendritic spines [47-49], where neurons try to control microglia activation in HD [50]. mHTT-expressing neural progenitor cells secrete vesicles containing mHTT oligomers to trigger striatal neurodegeneration [51], where mHTT oligomers also disrupt energy metabolism of neighboring cells [52]. Not only mHTT affects brain metabo-lism but it also causes the dysfunction of other organs such as liver [53]. In the below sec-tions, we discuss how mHTT affects astrocytes and oligodendrocytes in the neuropatho-genesis of HD.”

  1. Page 7, Conclusion:Please present conclusion with future perspective, including weaknesses or limitation in the present review, potentials, the ultimate goal, research or knowledge needed to achieve, the biggest challenge in this goal, and future research direction, among others.

Response: As the reviewer suggested, we added a paragraph regarding the future perspective at the end of the Conclusion section as shown below (lines 459-483):  

“The scope of this review is to briefly introduce the previous and recent studies about mechanisms of HD pathologies and therapeutic strategies and that our review has limita-tion in the scope. Just in case, for the readers need further information, we recommend the previous reviews dealt with the specific aspects of striatal vulnerability [249], white matter phenotype [250], cerebellar dysfunction [251], progression of cell type-specific phenotype [252], microglial activation [253], synapse [254], intracellular transmission of mHTT [255], protein-protein interactions [256], biochemical alterations and HTT dynamics [257,258], posttranslational modifications [259], proteostasis [260], autophagy [261,262], redox ho-meostasis [263], metabolism [264,265], HTT mRNA [266,267], Ca2+ and dopamine signal-ing [61], inflammation [268], in vitro modelling of HD [269,270], striatal neurogenesis [271], stem cell treatment [272-280], electric stimulation therapy [281], network connectiv-ity in presymptomatic HD brain [282], non-motor symptoms [283], gut microbiome [284], human immunodeficiency virus [285], diagnosis [286,287], clinical progression [288], treatment for the symptoms [289], physical therapy [290], psychological interventions [291,292], and management of agitation [293]. Collectively, the previous studies have po-tential to reveal spatiotemporal and cell-type specific mechanism of HD pathology. The future challenges in HD research are brought by the complexity of the pathology from bi-ochemical level [294-305] to system level [306-310]. Accordingly, the ultimate mechanisms of HD pathology can be further scrutinized by state-of-the art research methods such as multi-omics approach combining transcriptome, proteome, and interactome [311], big data analysis with machine learning [312], and meta-analysis combining the publicly available data [313]. On the other hand, the potential HD therapeutics should specifically modulate the function of the striatal neurons while they prevent the adverse behavior of glial cells. High-throughput in silico and in vitro screening of chemical libraries [314-318] are expected to expedite the designing of beneficial compounds for HD.”

The manuscript contains five figures, two tables, and 177 references. The manuscript carries important value exploring the mechanism of mHTT neurotoxicity and potential intervention in HD. I recommend this manuscript for publication after minor revision.

Reviewer 3 Report

In the manuscript, the authors discussed how mHTT protein disrupts the function of medium spiny neurons modulates epigenetic modifications and transcriptional pathways. In addition, the authors defined how non-cell-autonomous pathways lead to damage and death of MSNs under HD pathological conditions and overviewed the therapeutic approaches for HD.

 Although the article is generally well written and covers main topics, there are issues, the authors need to consider.

  1. In the introduction section, the authors should :
  • mentioned that HTT protein is important in embryonal development (full HTT KO is lethal), neuronal function (work 11 from suggested titles), or cardiomyocyte development and metabolism (works 8,9,4 or 2)
  • added the information that HTT is expressed not only in the brain but also in other organs such as heart or skeletal muscle, also mHTT inclusions are found for example in HD affected skeletal muscle,
  • mentioned the nowadays HD is considered also a multi-system disorder mainly due to skeletal muscle and heart function derangements, in those systems energy metabolism and mitochondrial changes were also noted.
  1. In the main sections, the authors are very much welcome to consider changing the order of the main paragraphs: 1. Epigenetic modification in HD, 2. Role of HTT in vescile trafficing  and 3. Mitochondrial dysfunction in HD. Moreover, authors need to consider discussed other pathways in which WT HTT is involved such as bioenergetics, purines metabolism, autophagy.
  2. Due to the fact, that authors discussed also the PGC-1 alpha role in HD pathophysiology, they may add to Table 2 (Metabolism and Mitochondrial dysfunction) also PPAR agonists (Dickey, A., Pineda, V., Tsunemi, T. et al. PPAR-δ is repressed in Huntington's disease, is required for normal neuronal function and can be targeted therapeutically. Nat Med 22, 37–45 (2016))
  3. Furthermore, in the conclusion section authors should also consider and discussed that all observed disruption in CNS and other organs in HD could be caused not also by mHTT accumulation but also by the loss of functionality of HTT protein which is also very important in HD pathophysiology. 

Author Response

Response to Reviewer 3:

Thanks for the constructive feedback. We went through each comment carefully, revised the manuscript and responded point by point.

In the manuscript, the authors discussed how mHTT protein disrupts the function of medium spiny neurons modulates epigenetic modifications and transcriptional pathways. In addition, the authors defined how non-cell-autonomous pathways lead to damage and death of MSNs under HD pathological conditions and overviewed the therapeutic approaches for HD.

 Although the article is generally well written and covers main topics, there are issues, the authors need to consider.

  1. In the introduction section, the authors should :
  • mentioned that HTT protein is important in embryonal development (full HTT KO is lethal), neuronal function (work 11 from suggested titles), or cardiomyocyte development and metabolism (works 8,9,4 or 2)

Response: As the reviewer suggested, the following sentences and references were added to the first paragraph of introduction section (lines 35-37).

“HTT has a crucial role in embryonic development by showing the death of embryos of huntingtin homozygous knockout mice by day 7.5 [5]. On the other hand, HTT plays an important role in cardiomyocytes cellular energy and nucleotides metabolism [6].”

  • added the information that HTT is expressed not only in the brain but also in other organs such as heart or skeletal muscle, also mHTT inclusions are found for example in HD affected skeletal muscle,

Response: Regarding the suggestion of Reviewer C, we added the information and references about expression of HTT in Skeletal muscle and Heart as follows (lines 40-43):

“Although most of studies show intracellular inclusions formed by mutant HTT (mHTT) protein in selected regions of the brain such as the striatum and cortex of HD brain [8,9], other studies find the expression and function of mHTT in skeletal muscle and heart as well [10].”

  • mentioned the nowadays HD is considered also a multi-system disorder mainly due to skeletal muscle and heart function derangements, in those systems energy metabolism and mitochondrial changes were also noted.

Response: We added following sentences to the introduction as the reviewer requested (lines 38-43):

“Currently, HD is considered as a multi-systemic neurodegenerative disease due to skeletal muscle and heart function disturbances with energy metabolism and mitochon-drial alterations beyond the brain dysfunction [7]. Although most of studies show intra-cellular inclusions formed by mutant HTT (mHTT) protein in selected regions of the brain such as the striatum and cortex of HD brain [8,9], other studies find the expression and function of mHTT in skeletal muscle and heart as well [10].”

  1. In the main sections, the authors are very much welcome to consider changing the order of the main paragraphs: 1. Epigenetic modification in HD, 2. Role of HTT in vescile trafficing and 3. Mitochondrial dysfunction in HD. Moreover, authors need to consider discussed other pathways in which WT HTT is involved such as bioenergetics, purines metabolism, autophagy.

Response: Thank you for your kind advice. We reordered the main paragraphs as the reviewer suggested. Also, we added the following paragraph in section 4 (lines 276-285), and section 5 (lines 322-324):

“In addition, wtHTT is also involved in autophagy as reviewed in [146]. wtHTT form complex with p62 to enhance cargo recognition, where depletion of wtHTT results in empty autophagosome [147]. C-terminal domain of wtHTT has structural homology with yeast autophagy scaffold protein 11 and both proteins show similar protein-protein inter-action patterns [148]. Interestingly, deletion of N-terminal domain of wtHTT in mouse suffers from DNA damage in striatum and cortex without any difference in autophagy function [149]. wtHTT is also associated with ER, where ER stress release the wtHTT to promote autophagy (reviewed in [150]). Also, wtHTT has important role in homeostasis of presynaptic and postsynaptic terminal [151]. Loss of wtHTT lead to dysfunction of synap-tic vesicle endocytosis in striatal neurons [152].”

“Moreover, the previous studies demonstrated that down regulation of wtHTT is related to mitochondria dysfunction by inability of the mitochondria to generate ATP [172] and diminished purines and inosine monophosphate [173].”

  1. Due to the fact, that authors discussed also the PGC-1 alpha role in HD pathophysiology, they may add to Table 2 (Metabolism and Mitochondrial dysfunction) also PPAR agonists (Dickey, A., Pineda, V., Tsunemi, T. et al.PPAR-δ is repressed in Huntington’s disease, is required for normal neuronal function and can be targeted therapeutically. Nat Med 22, 37–45 (2016))

Response: As the reviewer suggested, we added KD3010 which is PPAR agonists as transaction to Table 2 (line 436):

Transactivation

KD3010

Increased PPARδ transactivation

Transgenic mouse HD Models

[247]

Mitochondrial dysfunction

PGC-1α

Upregulate mitochondrial gene

[223]

  1. Furthermore, in the conclusion section authors should also consider and discussed that all observed disruption in CNS and other organs in HD could be caused not also by mHTT accumulation but also by the loss of functionality of HTT protein which is also very important in HD pathophysiology. 

Response: As the reviewer suggested, we added the following sentence to the conclusion (lines 286-296):

“We need to provide attention to an important HD pathophysiology that the dysfunc-tions of central nervous system and other organs in HD are caused by mHTT accumula-tion as well as by the loss of functionality of wtHTT protein. Molecular simulation reveals that mHTT oligomer also sequester wtHTT [153]. Indeed, wtHTT protein expression level is inversely correlated to the age of onset [154]. In macrophage, reduced wtHTT level is associated with decreased cytokine and increased phagocytosis [155]. Research is ongoing to reveal the wtHTT function and structure further. RNA-seq of wtHTT knockout neural cell shows that wtHTT has a role in development of neurons and neurotransmission [156]. Cryo-electron microscopy structure of wtHTT confirms its role in protein-protein in-teraction [157]. The importance of the loss of functionality of wtHTT is associated with clinical safety of HTT gene therapy as reviewed previously [158].”

Reviewer 4 Report

The manuscript “Non-cell autonomous and epigenetic mechanisms of Huntington’s Disease” by Kim and colleagues is interesting and comprehensive. I only have a few minor concerns:

I suggest an English revision as some phrasing is incorrect, some examples just in the few first lines include:

- “Especially”, line 14, is incorrect (should be “specifically” or something similar)

- “the higher CAG repeats” line 31 (should it say something like “the high number of…”?)

- Line 46 “Can be occurred”?

The reference to the Figures is missing in the text

There is some random highlighting in the text which should be removed

It would be interesting if the authors could integrate section 3 with commenting/discussing evidence on the role of RNA-based epigenetic alterations (e.g. miRNA, lncRNAs etc)

I would re-arrange the structure of the manuscript for consistency: for example, if the authors mention they will discuss other cell types first lines 52-53), then this should be the first paragraph. Moreover, cellular mechanisms such as vesicular trafficking and mitochondria dysfunction should be discussed one after the other

Author Response

Response to Reviewer 4:

Thanks for the valuable comments. We went through each comment carefully, revised the manuscript and responded point by point.

The manuscript “Non-cell autonomous and epigenetic mechanisms of Huntington’s Disease” by Kim and colleagues is interesting and comprehensive. I only have a few minor concerns:

  1. It would be interesting if the authors could integrate section 3 with commenting/discussing evidence on the role of RNA-based epigenetic alterations (e.g. miRNA, lncRNAs etc)

Response: As reviewer suggested, we added one paragraph to the end of section 3 (Epigenetic modifications in the pathogenesis of HD) to discuss about the role of miRNAs in HD as follows (line 200-223):

“Our group has found that SET Domain Bifurcated Histone Lysine Methyltransferase 1 (SETDB1/ESET), a histone H3 at lysine 9 (H3K9)-specific methyltransferase, is elevated in the striatal neurons of HD patients and HD transgenic (R6/2) mice [115]. In parallel, the level of histone H3K9me3 is increased in the striatal neurons of HD patients and in HD transgenic (R6/2) mice. This study has proven that the SETDB1-H3K9me3 pathway is in-volved in silencing of genes in HD. Interestingly, not only SETDB1 modulates the nuclear gene transcription though heterochromatin remodeling, but it also down regulates the nu-cleolar gene transcription (ribosomal RNA components) by increasing methylation of UBF1 (upstream binding protein 1). SETDB1 interacts with UBF1 and trimethylates at ly-sine-232/235 in the nucleolus of striatal cells. As a result, trimethylated UBF1 leads to nu-cleolar chromatin condensation and down regulates the transcription of ribosomal DNA (rDNA) [12]. This study presents a novel epigenetic mechanism that SETDB1-UBF1 tri-methylation pathway is associated the nucleolar chromatin remodeling and dysfunction of rDNA transcription in the pathogenesis of HD.

Moreover, several studies have focused on microRNAs (miRNAs) which are involved in the early differentiation, development, and function of neurons [116,117]. miR-146a is one of the major regulators of the NF-κB pathway which can also target human and mouse HTT gene [118,119]. Das et al. demonstrate that heat shock factor 1 is regulated by this miRNA, resulting suppression of mHTT aggregates in HD cells [120]. Another study confirmed that miR-214 directly targets the HTT gene which can suppress mHTT aggre-gation in an HD cell model [120,121]. On the other hand, Bucha et al. show the upregula-tion of miRNA-214 in HD cells could regulate mitofusin2, resulting alteration of mito-chondrial morphology [122]. Therefore, this miRNA can be considered as a critical node for therapeutic targets in HD pathogenesis.”

  1. I suggest an English revision as some phrasing is incorrect, some examples just in the few first lines include:
  • “Especially”, line 14, is incorrect (should be “specifically” or something similar)
  • “the higher CAG repeats” line 31 (should it say something like “the high number of…”?)
  • Line 46 “Can be occurred”?

Response: As reviewer suggested, we corrected the incorrect phrasings.

  1. The reference to the Figures is missing in the text

Response: As reviewer suggested, we added the references to the figures and tables.

  1. There is some random highlighting in the text which should be removed

Response: As reviewer suggested, we removed the highlightings in the text.

  1. I would re-arrange the structure of the manuscript for consistency: for example, if the authors mention they will discuss other cell types first lines 52-53), then this should be the first paragraph. Moreover, cellular mechanisms such as vesicular trafficking and mitochondria dysfunction should be discussed one after the other. 

Response: As reviewer suggested, we re-arranged the structure of the manuscript starting from “non-cell autonomous cell death pathway in HD section”, followed by “Epigenetic modifications in the pathogenesis of HD”, “Roles of wild type HTT (wtHTT) versus mHTT in vesicle trafficking”, and “Mitochondria dysfunction in HD”.